# Behçet’s Disease: A Radiological Review of Vascular and Parenchymal Pulmonary Involvement

**DOI:** 10.3390/diagnostics12112868

**Published:** 2022-11-19

**Authors:** Caterina Giannessi, Olga Smorchkova, Diletta Cozzi, Giulia Zantonelli, Elena Bertelli, Chiara Moroni, Edoardo Cavigli, Vittorio Miele

**Affiliations:** Department of Emergency Radiology, Careggi University Hospital, 50134 Florence, Italy

**Keywords:** Behcet’s disease, lung diseases, pulmonary arteries diseases, thrombosis

## Abstract

Behcet’s disease (BD) is a chronic systemic inflammatory disorder characterized by underlying chronic vasculitis of both large- and small-caliber vessels. Thoracic involvement in BD can occur with various types of manifestations, which can be detected with contrast-enhanced MSCT scanning. In addition, MR can be useful in diagnosis. Characteristic features are aneurysms of the pulmonary arteries that can cause severe hemoptysis and SVC thrombosis that manifests as SVC syndrome. Other manifestations are aortic and bronchial artery aneurysms, alveolar hemorrhage, pulmonary infarction, and rarely pleural effusion. Achieving the right diagnosis of these manifestations is important for setting the correct therapy and improving the patient’s outcome.

## 1. Introduction

Behcet’s disease (BD) is a chronic systemic inflammatory disorder characterized by underlying chronic vasculitis, which is an inflammatory process involving blood vessels. Both large- and small-caliber vessels are involved in BD, so the disease is classified according to Revised International Chapel Hill Consensus Conference Nomenclature of Vasculitides in the subgroup of “variable vessel vasculitis” [1]. The etiology is largely unknown, although autoimmune and inflammation-induced processes are certainly involved. The pathogenesis is probably related to environmental factors acting on a genetic predisposition, linked to various epigenetic modifications (for example, factor Human leucocyte antigen B51–HLA-B51 mutations) [2,3,4,5]. It was traditionally called “the Silk Road Disease”, due to its high prevalence in Turkey and Iran [6]. Nowadays, due to migration flows, the disease has also spread to Europe and the USA, with a prevalence of 10.3/100,000 people [7]. According to Gulen, H., et al., the clinical manifestations in Western countries seem to be more severe, but this is probably due to a delay in the diagnosis and treatment of the mild forms in areas where physicians are less familiar with BD [8].

Because there are no specific diagnostic laboratory tests or histopathologic findings, the diagnosis of BD is often challenging. It relies on clinical criteria and often takes several years to establish a definitive diagnosis after the appearance of the initial manifestations [9]. The clinical course of BD usually follows a relapsing–remitting course with heterogeneous clinical manifestations. Disease activity after the first years tends to decrease, leading to complete remission within 20 years in around 60% of patients [10].

Although BD can be seen at any age, the mean age of onset is mainly between the ages of 20 and 40. In most recent reports, males are more involved than females [11]. Different male-to-female ratios have been recorded in some countries; there is a male predominance in Middle Eastern countries, while a female predominance is seen in the USA, Korea, and UK [12].

As above mentioned, the most used classification criteria are the International Study Group on Behcet’s disease criteria (ISG) published in 1990 and the International Criteria for Behcet’s Disease (ICBD) published in 2006 and revised in 2010. The previous ISG criteria for Behcet’s have excellent specificity but lack sensitivity [13]. The revised ICBD criteria demonstrate a sensitivity of 94.8% and acceptably high specificity (90.5%) [14,15].

As a multisystemic disease, clinical manifestations can involve nearly the whole body. The most common manifestations are ocular involvement (anterior uveitis, posterior uveitis, and retinal vasculitis), genital or oral aphthosis, skin lesions (pseudo-folliculitis, skin aphthosis, and erythema nodosum), and neurological and vascular manifestations (arterial thrombosis, large vein thrombosis, and phlebitis). Some manifestations are rarer, such as joint involvement, gastrointestinal manifestations (6.3%), epididymitis (7.2%), and pleuropulmonary (1.8%) and cardiac manifestations (1.8%) [16,17].

Diagnostic imaging is a fundamental step in evaluating the patient with suspected BD, especially to assess complications in the emergency department. The widespread use of computed tomography pulmonary angiography (CTPA), with its rapid execution, often allows the identification of the cause of the massive hemoptysis, which could be the first manifestation of the disease in the emergency room. In case of hemodynamically stable patients, vasculitic involvement can also be studied with magnetic resonance imaging (MRI), with or without the injection of a gadolinium-based contrast medium. This article focuses on the lung vascular system involvement in BD and the main radiological findings related to the pulmonary, pleural, and mediastinal extension of the disease.

## 2. Imaging Approach to Thoracic Involvement in Behcet’s Disease

Thoracic involvement in BD is reported in about 1–10%; however, the prevalence depends on the geographical area, with a peak in Egypt (41% of BD cases) [18,19]. Major vascular involvement has a definite male preponderance and is usually an early manifestation [20]. Several authors agree that one of the worst complications of BD is the vasculitic involvement of the small and large pulmonary arteries, which presents with aneurysms and thrombosis. Systemic arteries (aorta and bronchial arteries) and both pulmonary and systemic veins can also be affected. A typical feature of BD is thrombosis of the superior vena cava (SVC) accompanying fibrosing mediastinitis. Parenchymal phenomena such as pneumonia or alveolar hemorrhage may occur [21].

First-line imaging is based on chest X-ray (CRX), which may be normal or showing indirect signs of vasculitis. Pulmonary artery aneurysms present as hilar enlargement or as round opacities, while mediastinal widening can suggest an aneurysm of the thoracic aorta [18,22]. CRX can be used as a screening method for detecting thoracic involvement or in a follow-up program to assess the therapeutic response. The best imaging method to detect all thoracic manifestations of BD is CTPA, which accurately shows arterial aneurysms thanks to multiplanar vascular reconstruction [18,23]. CTPA can detect all the main features of BD, especially for aneurysms, showing a relationship with surrounding structures and giving indications for surgical approach [19,24,25,26]. Zhou, J., et al. reported that spectral CT is better in detecting thrombosis at a low energy level of 40 keV using virtual monoenergetic imaging than conventional imaging at 120 kV [27]. Angiography is not recommended in patients with BD because of the increased risk of aneurysm formation at the puncture site and venous thrombosis after the injection of the contrast material [18]. However, angiography can be used as a lifesaving method in cases of massive hemoptysis [28]. Open surgery is an option in the case of non-responding to medical treatment, but the endovascular technique is recommended especially for patients with a high surgical risk [29]. MRI can detect aneurysms of the aorta and pulmonary arteries (Figure 1 and Figure 2). Despite MRI being less sensitive than CTPA in identifying small lung vascular aneurysms, the advantages of MRI are connected to its use in allergic patients or those with renal failure, through the use of sequences where blood flow can be enhanced without contrast material injection [30,31].

### 2.1. Vascular Involvement

#### 2.1.1. Systemic Arterial Manifestations

In BD, arterial pathology occurs in 7 to 23% of cases and is more common than the venous one [32]. Aneurysm formation is much more frequent than arterial thrombosis. The most involved sites are the abdominal aorta and the iliac and femoral arteries, while in the thorax, they are the aortic arch and the coronary and subclavian arteries (Figure 1 and Figure 2) [21,33]. The pathogenesis is linked to inflammatory phenomena affecting the vasa vasorum supplying the middle tunica and the adventitia; then, the arterial wall weakens and dilates, forming saccular aneurysms [34]. The rupture of the aneurysms can lead to sudden death from a massive hemorrhage [35]. More rare manifestations can be stenosis, arterial occlusion, and pseudoaneurysms, which can cause ischemic phenomena. Yazgan et al. reported that patients with BD and pulmonary artery involvement (PAI) show a larger diameter of the bronchial arteries than those without PAI. This is probably due to increased pressure in the pulmonary arteries (PAs) secondary to vasculitic PA phenomena. Hypertrophic bronchial arteries should be assessed upon CTPA scanning, especially in patients with hemoptysis that do not respond to immunosuppressive treatment [31].

#### 2.1.2. Pulmonary Arterial Manifestations

Pulmonary artery involvement (PAI) is a rare manifestation of BD with an overall incidence reported in less than 5% [36]. It usually occurs 3–4 years after disease onset [37]. Pulmonary artery aneurysms (PAAs) are the most common form of pulmonary involvement in BD, followed by pulmonary artery thrombosis (PAT), pulmonary infarction, and pulmonary parenchymal anomalies. In the case of the detection of PAAs, BD should always be suspected first. PAAs are the second most common site of arterial involvement, preceded by the abdominal aorta [38]. Pulmonary aneurysms present as saccular or fusiform dilatations that show homogeneous contrast filling simultaneously with the pulmonary artery. PAAs in BD predominantly affect the right lower lobar artery, followed by the right and left main pulmonary arteries (Figure 3 and Figure 4). They are multiple, usually pseudoaneurysms, with various diameters and in situ thrombosis [21]. The most frequent symptom is massive hemoptysis caused by aneurysm rupture with erosion into a bronchus [39]. Pulmonary aneurysms are classified using CTPA into six radiologic patterns: aneurysmatic wall enhancement on post-contrast CTPA; true “stable” pulmonary artery aneurysms (PAAs) or bronchial artery aneurysm, characterized by adherent in situ thrombosis; “unstable”, leaking PAA; stable or unstable pulmonary artery pseudoaneurysms (PAPs) with loss of aneurysmal wall definition (most prone to rupture); and right ventricular strain (RVS) with or without intra-cardiac thrombosis [40]. It is important to differentiate in situ thrombosis within the lumen of the aneurysm seen in true PAAs from marginal extraluminal thrombosis in PAPs, as this indicates a chronic leak through the inflamed aneurysm wall, which predisposes to rupture. It is also mandatory to highlight at CT any connection between PAAs and the adjacent bronchus [33].

Clinically silent aneurysms are often accidentally detected on CXRs or CT scans as hilar enlargement or round, lobulated opacities. Screening for silent aneurysms is recommended whenever anticoagulation therapy is considered in patients with confirmed BD, to avoid the risk of hemorrhages [41]. CTPA has largely replaced angiography as the tool for the diagnosis of PAAs. MRI is less commonly used to detect PPAs, because it appears to be less sensitive than CT for the diagnosis of small aneurysms [42]. Trans-thoracic echocardiography (TTE) may be useful; the wall of the pulmonary arteries at TTE is thicker in individuals with BD who have major organ involvement than in those with only mucocutaneous symptoms, which is an index of disease severity [43,44,45,46,47,48,49].

PAT may be isolated (33%) or coexist with PAAs in 25% of cases. Both in situ thrombosis and embolization are possible mechanisms of pulmonary artery thromboembolism; according to Seyahi, the mechanism of in situ-thrombosis is observed rather than embolism. Lower-lobe arteries are mainly involved in both cases of PAI (PAAs and PAT), the number of vessels involved is significantly lower in patients with isolated PAT. These have similar clinical features to those with PAAs, although massive hemoptysis is observed less frequently in patients with isolated PAT [50,51]. In the case of isolated PAT, CXR could be normal; thus CT scanning is an optimal noninvasive imaging modality for initial and follow-up evaluations [35,52,53]. In situ thrombosis is caused by various thrombophilic conditions related to BD, with the first being factor V Leiden deficiency, which makes it difficult to choose anticoagulant therapy, as it may increase the risk of hemoptysis [54].

The combination of peripheral thrombophlebitis and PAAs has been called “incomplete Behcet’s disease” (Hughes–Stovin syndrome (HSS)) due to the similarities between the imaging and pathological findings [55]. Patients with HSS may have PA aneurysms and thrombophlebitis with or without oral or genital ulcerations. In rare cases, bronchial artery aneurysms or recurrent pulmonary embolism could also be seen [56]. Typical manifestations of this syndrome are cough, shortness of breath, fever, and chest pain [57]. As BD, HSS may present with hemoptysis [58]. As for PAAs, steroid therapy associated or not with immunosuppressants has been suggested [59,60].

#### 2.1.3. Systemic Veins

The vascular autoinflammatory phenomena of BD also involve the veins and manifest themselves predominantly with thrombosis. Venous thrombosis occurs in a range of 15 to 45% of affected patients, predominantly as superficial and deep vein thrombosis. Various studies have affirmed that up to 30% of cases may occur as major venous thrombosis (superior and inferior vena cava (SVC and IVC), portal vein, hepatic vein, and dural sinuses) [61,62]. The pathogenesis of thrombosis in BD is due to both vasculitic inflammatory phenomena of the large veins and hypercoagulability status [63].

In the chest, thrombosis begins in the SVC and sometimes involves the adjacent veins or the right ventricle (Figure 5 and Figure 6). In the chronic phase, obliteration of the vein with the formation of collateral circles may occur. Usually, the re-inhabiting collateral pathways converge at the azygos or hemiazygos veins; however, cases of BD and SVC syndrome with hepatic collateral circles have also been described [64]. CRX may show indirect signs, such as mediastinal widening. On CT scans, SVC thrombosis appears as narrowing, lack of opacification, wall thickening, and filling defect, accompanied by oedema of the surrounding soft tissue; moreover, multiple collateral vessels are usually present in the area drained by the SVC [35,65]. MRI, with gradient-echo or spin-echo cine-sequences, can detect the involvement of mediastinal veins and establish their extension [66]. Thrombosis clinically presents as SVC syndrome, with dyspnea, jugular turgor, facial oedema, and swollen collateral veins on the front chest wall. Calamia et al. reported that BD is the most common cause of SVC syndrome in countries where it is endemic [67]. SVC thrombosis in BD can be associated with several symptoms and manifestations, such as iliac vein thrombosis or cardiomyopathy. A case report by Tadeu Ferreira de Paiva Jr. described an SVC syndrome without evidence of thrombosis; the lumen reduction was due to a thickening of the vessel wall, probably caused by inflammatory phenomena [4,68,69,70]. In another case reported by Harman, an SVC thrombosis was due to extrinsic compression by mediastinal fibrosis [71]. SVC obstruction without thrombosis can be due also to a large saccular aneurysm of the subclavian artery, as described by Nair et al. [72,73].

### 2.2. Pleural and Lung Parenchymal Involvement

The spectrum of lung parenchymal findings may be confused with other common parenchymal diseases. Parenchymal lesions in BD could be isolated or may be seen frequently in cases of PAI [74]. Pulmonary vasculitis and thrombosis of pulmonary vessels result in infarction, hemorrhage, and focal atelectasis [75]. Normally, parenchymal lesions are subpleural, wedge-shaped, or ill-defined increased-density opacities that are considered focal vasculitis with hemorrhage, infarction, and inflammation (Figure 7). In chronic pulmonary thromboembolism, damaged lung tissue can be replaced by fibrosis or emphysema. The most common cause of mosaic perfusion defects on CT scans is a small airways inflammation and fibrosis with focal or diffuse airway narrowing and air trapping. Lung parenchyma mosaic perfusion defects also may reflect the status of chronic pulmonary artery occlusion [76].

Rarely, parenchymal involvement may manifest as bronchiolitis obliterans, organizing pneumonia, eosinophilic pneumonia, diffuse alveolar hemorrhage, and interstitial lung disease [77]. Additionally, BD may cause tracheal and proximal airway occlusions due to scars, which may lead to luminal stenosis [78,79]. Pneumonia in BD can be the result of the inflammation of pulmonary parenchymal vessels or may represent a complication of immunosuppressive therapy [80]. Chebbi et al. recently reported a case of a patient with history of multiple PAAs presenting with massive hemoptysis; the CT scan showed a regular-shaped, thin-walled cavity with an air and fluid level, which connected to a segmental bronchus. Apparently, it appeared to be a cavitated lesion of the parenchyma, but after surgery, it was shown to be a thrombosed PAA with bronchial fistula [81]. Conventional CXR is usually used for the initial evaluation of pulmonary involvement. High-resolution CT (HRCT) scans are the best radiological tool for the evaluation of pulmonary pathological features. Chest HRCT is non-invasive and provides excellent delineation of the parenchymal lesions [82]. Pleural vasculitis can lead to the formation of pleural nodules, which may appear as parenchymal subpleural lesions. Pleural effusion as a manifestation of BD is rare and is due to SVC thrombosis or pulmonary infarction in most cases [35,83]. Chylothorax, chylopericardium, and ascites can occur as a complication of SVC syndrome in BD [84].

### 2.3. Mediastinum and Heart

Mediastinal lymphadenopathy in BD is reported as a probable reaction to a chronic inflammatory process. Pericardial effusion may result from chronic pericardial inflammation or SVC thrombosis. Cardiac involvement of BD may occur in the form of intracardiac thrombosis (ICT), endocarditis, myocarditis, pericarditis, endomyocardial fibrosis, coronary artery disease, myocardial infarction, aneurysm of the sinus of Valsalva, periaortic pseudoaneurysm, and valvular and conduction system anomalies [35,85]. The literature describes a discrepancy between the types of cardiac complications among the patient cohorts in different geographical regions and ethnic groups. Recent epidemiological studies confirmed that the most frequent type of cardiac manifestations of BD in Turkey, the Middle East, and the Mediterranean regions is ICT, while aortic valve regurgitation remains restricted to the Asian countries [86]. A recent Iranian study of 7650 patients reported about 2% of ICT [87]. This cardiac complication of BD was described as a finding mainly associated with PAAs. The right side of the heart is the most frequent site of involvement, with the ventricle being more affected than the atrium. ICT is more commonly seen in young male patients and usually appears soon after disease onset [88]. The acute symptoms are usually facial swelling, dyspnea, fever, hemoptysis, and palpitation [86,89]. ICT is frequently associated with pulmonary arterial or venous thrombosis and endomyocardial fibrosis; therefore, it is difficult to demonstrate whether the thrombi are secondary to these conditions or are caused by a de novo process [51]. CTPA and MRI can show a filling defect in the ventricle and can provide additional information, such as the presence of pulmonary arterial thromboembolism or a lung parenchymal lesion. Aneurysms of the Valsalva sinus, if untreated, may lead to complications such as RVS, coronary artery occlusion, aortic regurgitation, and congestive heart failure [90]. Coronary artery pseudoaneurysm and giant coronary artery aneurysm can be found [91,92,93,94]. Recently, a case of BD with myocarditis was reported [95]; in this case, transesophageal echocardiography is more useful than TTE for making this diagnosis. CT or MR angiography is safer than conventional angiography for analyzing the status of vascular involvement, because an arterial or venous catheter can induce either thrombosis or aneurysm formation [35].

## 3. Management and Therapy

The clinical management of BD patients is multidisciplinary, involving clinicians, pulmonologists, and cardio-vascular surgeons. Imaging is fundamental for detecting pulmonary and vascular involvement, especially in the acute phase; moreover, in patients with chronic symptoms, CXR, CT, and MRI exams could also be routinely performed to evaluate the evolution of the inflammatory disease. Patients are treated with medical treatment first, but open surgery is an option in non-responding cases, while the endovascular technique is recommended especially for patients at a high surgical risk [29]. As already mentioned above, PPAs are characterized by significant mortality and poor short-term survival in the early years [44]. The prognosis has been improved by earlier diagnosis and the introduction of immunosuppressive therapies, which should be the first-line treatment of choice. The EULAR 2018 recommendations strongly suggest the use of high-dose steroids in association with cyclophosphamide in PPAs, while the use of anti-TNF-α should be considered for refractory cases [45]. The use of anticoagulants in this condition is negligible and should be added to steroid treatment in case of venous thrombosis if a coexisting pulmonary aneurysm is ruled out. According to some studies, immunosuppressive drugs and steroid treatment may regress aneurysms in up to 75% of patients with PAAs [46,47]. Vascular surgery may be necessary to treat aneurysms in patients with rapidly expanding or recurrent PAAs, although 24% of these surgeries are complicated by graft occlusion and 13% by anastomotic pseudoaneurysm [48]. Endovascular treatment is the most preferred approach in subjects unresponsive to conservative treatment. The Amplatzer duct-occluder is currently the most used device for the management of large aneurysms. Interventional embolization is associated with higher risks of recurrence relapse (40% at five years) and reintervention for PAAs. Moreover, patients after lobectomy and decortication exhibit the highest mortality rates [49]. As for PAAs, in patients diagnosed with HSS, steroid therapy associated or not with immunosuppressants has been suggested [59]. However, in rare cases, pulmonary aneurysms may progress despite medical and surgical management, eventually requiring lung transplantation [60]. In case of systemic veins involvement, the treatment of thrombosis in atypical sites such as the vena cava is not based on anticoagulation alone but requires combined therapy with immunosuppressants and steroids [41]. Infliximab was recently found to be effective and well tolerated [73]. Finally, in the case of heart involvement, surgical treatment is controversial because of the high risk of recurrence and worse post-operative outcomes. Continuous steroid and immunosuppressive agent therapies are fundamental to reduce the risk of BD recurrence [90].

## 4. Conclusions

In conclusion, BD and HSS are pathologies with high mortality rates, but HRCT and CTPA findings are often non-specific; it is important to recognize and classify them, either through appropriate glossary terms or through a structured report, which can help both experienced and younger radiologists to improve diagnosis and patient management [96,97]. Especially in the emergency department, an accurate clinical evaluation is mandatory, together with diagnostic imaging, to reach a prompt diagnosis. As already demonstrated for lung cancer and acute infective pneumonia, deep learning and artificial intelligence technologies able to identify specific signs or biomarkers of vasculitis could also be useful tools in the future [98,99,100].

## Figures and Tables

**Figure 1 diagnostics-12-02868-f001:**
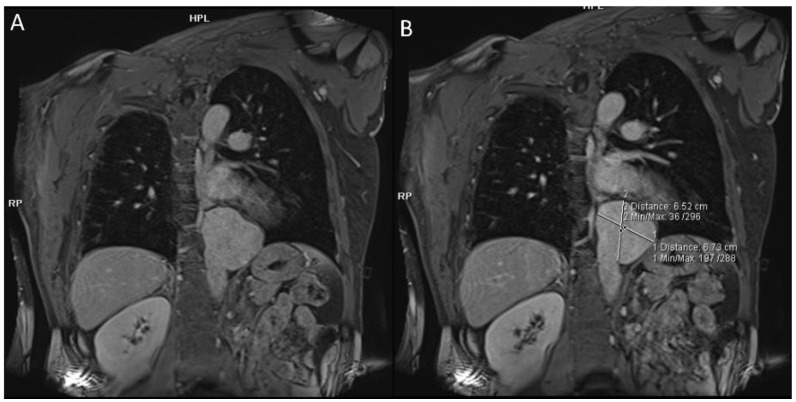
(**A**,**B**) Angio-MRI MIP reconstructions in BD patient that show aneurysm of thoracic aorta.

**Figure 2 diagnostics-12-02868-f002:**
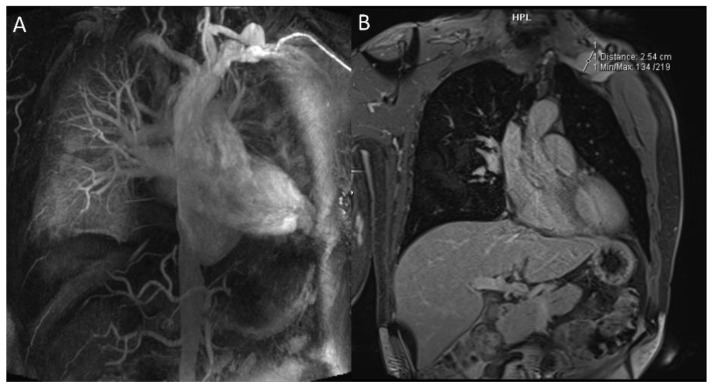
(**A**,**B**) Angio-MRI MIP reconstructions in BD patient that show aneurysm of left subclavian artery.

**Figure 3 diagnostics-12-02868-f003:**
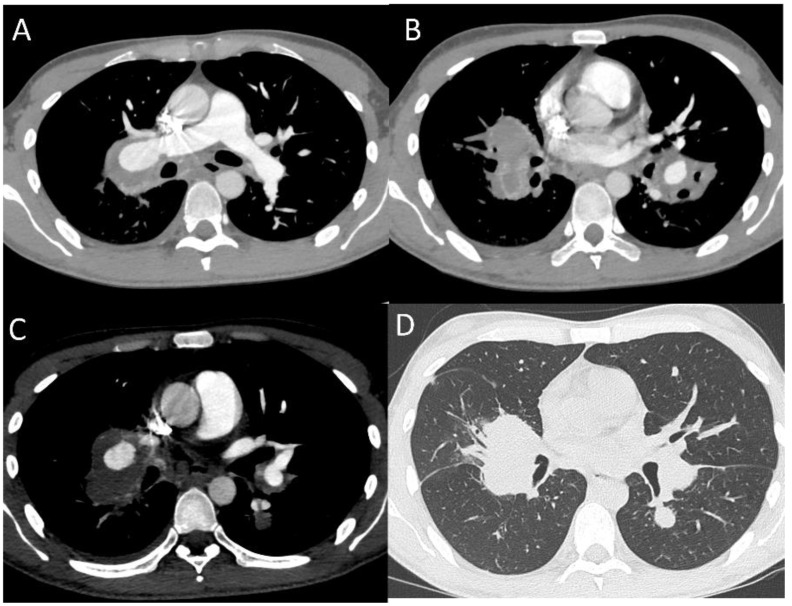
PAAs in BD patient: (**A**) Aneurysm of the middle lobar branch of the right pulmonary artery; (**B**,**C**) aneurysm of lower lobar branch of both pulmonary arteries; (**D**) parenchymal ground glass opacities beside the aneurysmatic artery branch.

**Figure 4 diagnostics-12-02868-f004:**
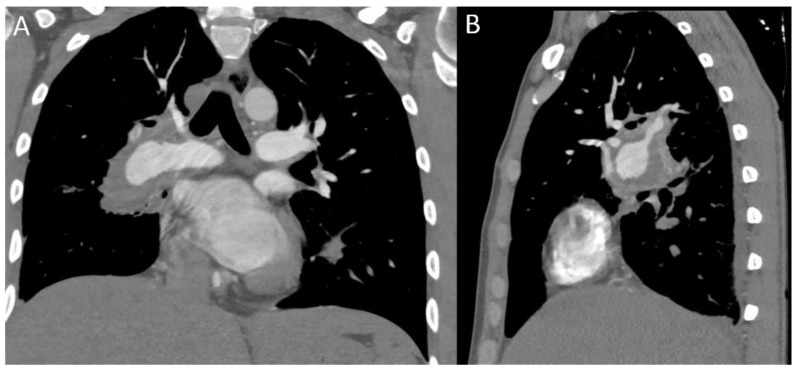
(**A**,**B**) Aneurysm of the middle lobar branch of the right pulmonary artery. MPR CT reconstructions in coronal and sagittal scans.

**Figure 5 diagnostics-12-02868-f005:**
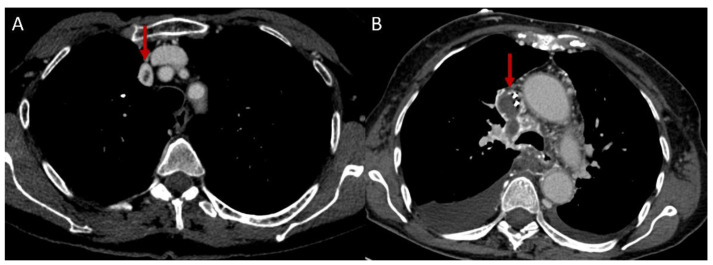
SVC thrombosis (red arrows): (**A**) minimum SVC thrombosis in BD patient; (**B**) SVC and azygos vein thrombosis in BD patient with CVC.

**Figure 6 diagnostics-12-02868-f006:**
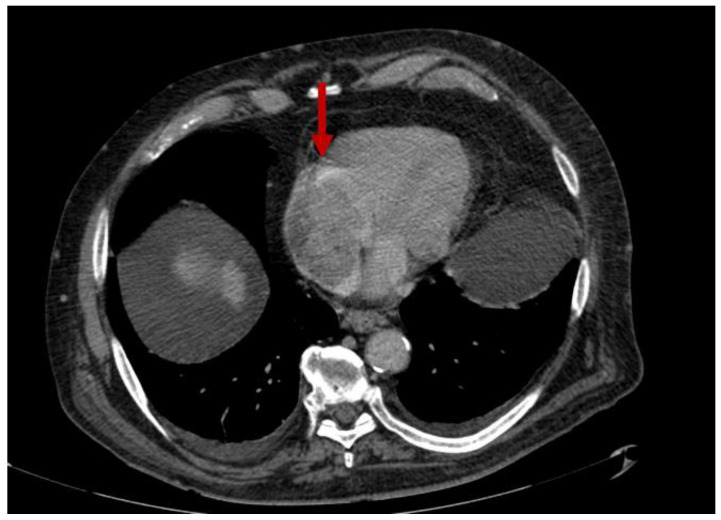
ICT in BD. The red arrow shows right-ventricle thrombosis.

**Figure 7 diagnostics-12-02868-f007:**
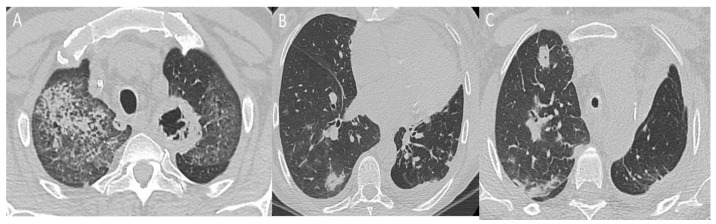
Parenchymal involvement in BD: (**A**) apical-site consolidation with cavitation associated with multiple ground-glass thickening indicating alveolar hemorrhage, a rare presentation of BD; (**B**,**C**) perivascular and sub-pleural consolidations and diffuse ground-glass opacities.

## Data Availability

Not applicable.

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
