# Peer review of "Behçet’s Disease: A Radiological Review of Vascular and Parenchymal Pulmonary Involvement"

_diagnostics, 2022, doi:10.3390/diagnostics12112868_

Round 1

Reviewer 1 Report

Abstract well written, concise enough. Clear description of the review scientific background, design and objectives.

Adequate explanation of the clinical characteristics of the disease.

Detailed description of the thoracic, vascular, parenchymal, pleural and cardiac involvement.

Innovative interpretation of the disease with updated bibliographic references and very detailed images that better clarify what is described.

The review therefore fully achieves its educational information objective.

Author Response

Thank you for the revision. We are all glad that this work satisfies your opinion.

Reviewer 2 Report

The authors provide a radiological view for Bechet’s disease. I have some doubts and hope authors could provide some clarifications.

1.     There are some typos and grammar errors in the writing. The manuscript is also not organized well. For example, there are duplicated sentences in abstract. Please spend time to revise the manuscript.

2.     There is lack of explanation on the legend of Figure 4.

3.     Please highlight the contributions of this paper. What are the core contributions?

4.     Please add more literatures in the introduction. The background is not clear for people to understand.

Author Response

Thank you for the revision. Here our answers to your comments.

  1. There are some typos and grammar errors in the writing. The manuscript is also not organized well. For example, there are duplicated sentences in abstract. Please spend time to revise the manuscript.

we have applied a consistent revision of the text, both structural and grammatical.

  1. There is lack of explanation on the legend of Figure 4.

we add an explanation of Figure 4.

  1. Please highlight the contributions of this paper. What are the core contributions?

We add the core contributions in the conclusions and in the text, while explaining the various involvement in Bechet disease.

  1. Please add more literatures in the introduction. The background is not clear for people to understand.

we clarify the introduction, not modifying the literature but with a better explaining of the clinical aspect of this disease.

Reviewer 3 Report

1. Significant grammatical changes needed as mentioned in the pdf comments

2. Introduction needs to be rewritten with respect to why this topic is chosen / significance of radiological findings of Bechets'

Author Response

Thank you for the revision. Here the answers for your suggestions.

  1. Significant grammatical changes needed as mentioned in the pdf comments

We have made an important grammatical and structural revision of the whole text.

2. Introduction needs to be rewritten with respect to why this topic is chosen / significance of radiological findings of Bechets'

We have partially re-written the introduction focusing on the main topic of the review.

Round 2

Reviewer 2 Report

 The manuscript is still not organized well. For example, the formats of some paragraphs are not correct. The citations of figures and references are not correct. Please continue improving it. Also, please continue checking the typos and grammar errors.    

Author Response

Thank you for your second review. We improve our review in language editing and in its structure. In our opinion, the paragraphs are well structured, we only improved language editing and added some observations in the introduction. Please specify in which way we could improve the review structure.

The citations of figures in the text are correct, and so are the references. Please specify your observation...

Reviewer 3 Report

Although it is a rarely written and important topic, I still feel that the paper could be improved with better writing.

1. Introduction needs to mention purpose of this review paper - why is imaging an important part in Bechets' disease

2. Grammatical errors to be corrected as mentioned. 

Author Response

Thank you for your comments.

  1. Introduction needs to mention purpose of this review paper - why is imaging an important part in Bechets' disease

We added some observations about the imaging approach and why radiological imaging is important in this disease.

2. Grammatical errors to be corrected as mentioned. 

We improved the text in language editing.

Best regards, 

the Authors.